# Early Embryo Exposure to Assisted Reproductive Manipulation Induced Subtle Changes in Liver Epigenetics with No Apparent Negative Health Consequences in Rabbit

**DOI:** 10.3390/ijms22189716

**Published:** 2021-09-08

**Authors:** Ximo García-Domínguez, Gianfranco Diretto, David S. Peñaranda, Sarah Frusciante, Victor García-Carpintero, Joaquín Cañizares, José S. Vicente, Francisco Marco-Jiménez

**Affiliations:** 1Laboratory of Biotechnology of Reproduction, Institute for Animal Science and Technology (ICTA), Universitat Politècnica de Valencia, 46022 Valencia, Spain; ximo.garciadominguez@gmail.com (X.G.-D.); dasncpea@upvnet.upv.es (D.S.P.); jvicent@dca.upv.es (J.S.V.); 2Casaccia Research Centre, Italian National Agency for New Technologies, Energy and Sustainable Development (ENEA), 00123 Rome, Italy; giandiretto@gmail.com (G.D.); sarah.frusciante@enea.it (S.F.); 3Institute for the Conservation and Breeding of Agricultural Biodiversity (COMAV-UPV), Universitat Politècnica de Valencia, 46022 Valencia, Spain; vicgarb4@upvnet.upv.es (V.G.-C.); jcanizares@upv.es (J.C.)

**Keywords:** assisted reproductive technologies, vitrification, stress, DNA methylation, metabolome, proteome, steroid biosynthesis

## Abstract

Embryo manipulation is a requisite step in assisted reproductive technology (ART). Therefore, it is of great necessity to appraise the safety of ART and investigate the long-term effect, including lipid metabolism, on ART-conceived offspring. Augmenting our ART rabbit model to investigate lipid metabolic outcomes in offspring longitudinally, we detected variations in hepatic DNA methylation ART offspring in the F3 generation for embryonic exposure (multiple ovulation, vitrification and embryo transfer). Through adult liver metabolomics and proteomics, we identified changes mainly related to lipid metabolism (e.g., polyunsaturated fatty acids, steroids, steroid hormone). We also found that DNA methylation analysis was linked to changes in lipid metabolism and apoptosis genes. Nevertheless, these differences did not apparently alter the general health status. Thus, our findings suggest that ART is likely to be a player in embryo epigenetic events related to hepatic homeostasis alteration in adulthood.

## 1. Introduction

During the preimplantation period, major epigenetic reprogramming occurs to provide the developing embryo with an epigenetic profile coherent with pluripotency, from which differentiated cells acquire lineage-specific transcriptional profile [1]. Nevertheless, modifications in environmental conditions can modify epigenetic reprogramming in the early embryo, changing its developmental programme [2]. The commonly reported “developmental plasticity” is thought to allow advantageous adaptive response mechanisms to be adapted appropriately to the environment in which the embryo will be developed [3]. However, stressful exposures outside their natural range in mammalian embryos that have not evolved appropriate mechanisms may result in non-adaptive responses [4]. This theory, now called developmental origins of health and disease (DOHaD), postulates that a suboptimal environment during embryo or fetal development may lead to adjustments in the anatomy, physiology and metabolism of various organ systems and thereby influence disease susceptibility [3,4].

A specific circumstance is that assisted reproductive technology (ART) handling occurs during preimplantation development, coinciding with global reprogramming of the epigenome and the establishment of epigenetic changes that persist into adulthood [2,5,6,7]. During gametogenesis and early embryogenesis, epigenome reprogramming occurs, resulting in the resetting of epigenetic changes and the conversion of differentiated gametes into totipotent embryos [8]. ARTs expose this sensitive window of embryo development to a synthetic [9,10], ex vivo environment, which might disrupt embryo development in ways that may have long-term consequences for health [9]. Therefore, the current epidemiological evidence and safety concerns regarding ART are a serious object of debate [5,11]. To date, it has been reported that infants conceived by ART have a 3-fold higher incidence of epigenetic disorders than infants conceived naturally [12]. However, knowledge regarding the long-term consequences of ART procedures remains relatively modest in humans, as most of the ART children are still young [9,13]. Furthermore, continuous changes in ART practice and numerous confounding factors associated with health complications (e.g., parental age, male or female infertility, social status, lifestyle) make it challenging to provide detailed answers from clinical outcomes [5]. Thus, published data are inconclusive, limited, and occasionally contradictory [14]. Investigating the effects of ART in fertile model species circumvents issues raised by underlying infertility and avoided other confounding factors, especially those regarding birth weight and growth trajectories, metabolic health, cardiovascular troubles and epigenetic alterations [15]. The mechanism underlying the alterations in offspring conceived with ARTs are not entirely clear, but epigenetic changes are a well-excepted contributor [2,5,14,15,16,17]. Here, we exploit our rabbit ARTs model to conduct a hepatic DNA methylation association study with metabolomics and proteomics in the F3 generation for embryonic exposure (multiple ovulation, vitrification and embryo transfer) conceived offspring, as well as assess the potential health implications.

## 2. Results

### 2.1. Normal Values for Hematological and Serum Metabolites Were Obtained for the F3 Generation Animals Ancestrally Exposed to Embryo Technologies

As shown in Table 1, no significant difference was found in the hematological profile (white blood cells, red blood cells, hemoglobin and hematocrit) between animals born from multiple ovulation, vitrification and embryo transfer (MOVET) and naturally conceived (NC) ones. However, attending to the serum biochemical data, lower levels of cholesterol and higher glucose levels were exhibited by MOVET animals (*p* < 0.05). Nevertheless, all hematology and biochemistry values correspond with seemingly healthy animals [18].

### 2.2. Liver Metabolite Profile Changes for the F3 Generation Animals Ancestrally Exposed to Embryo Technologies

First of all, we carried out an untargeted metabolomic analysis to gain a general overview of the metabolic changes occurring between the hepatic tissue of MOVET animals and NC animals. In total, 443 metabolites were quantified, of which 290 (65.5%) and 153 (34.5%) belonged from the semi-polar and non-polar fraction, respectively (Figure 1A). Specifically, of the total quantified metabolites, 211 (47.6%) were detected as differentially accumulated metabolites (DAMs), of which 160 (75.8%) and 51 (24.2%) belonged from the semi-polar and non-polar fraction, respectively (Figure 1A). The variability among these DAMs was investigated by building PCA diagrams, which clustered the samples according to their origin (MOVET or NC) both for the semi-polar (Figure 1B) and non-polar (Figure 1C) metabolites. HM clustering graphics showed an overall up-accumulation of the semi-polar DAMs (Figure 1D) in MOVET animals compared to the NC animals, whereas those non-polar were generally down-accumulated (Figure 1E).

### 2.3. Liver Protein Profile Changes for the F3 Generation Animals Ancestrally Exposed to Embryo Technologies

The complete spectral library included 28,685 of MS/MS spectra, corresponding to 14,737 distinct peptides and 1846 proteins with an FDR ≤ 1%. With the restrictions used for extraction parameters of the areas, 1491 proteins (FDR < 1%) were quantified in the eight samples. In comparing the proteomes between MOVET and NC animals, 97 DEPs were identified in mammalian taxonomy, of which 96 found their homologous identification in rabbit (*Oryctolagus cuniculus*) taxonomy. PCA (Figure 2A) and HM (Figure 2B) analysis showed that, despite expected individual variability, samples from each group were clustered together. From these DEPs, there was 52.1% downregulated (50/96) and 47.9% upregulated (46/96) in MOVET animals compared to NC ones. Pie charts representing the DEPs distribution according to their biological process (Figure 2C), molecular function (Figure 2D) and cellular component (Figure 2E) showed that most DEPs have catalytic and binding activity, are involved in cellular and metabolic process, and are located in cells and organelles.

Of the significantly different proteins, a total of 71 DEPs were recognized by the DAVID software (https://david.ncifcrf.gov), whose annotation and fold change has been described in Appendix A. Functional GO term enrichment and KEGG pathway analysis of DEPs were recorded in Appendix A. Thereby, 5 biological processes, 9 molecular function and 11 cellular component terms were enriched after functional analysis, suggesting disturbances in the immunological responses, long-chain fatty acid metabolism and cell cycle regulation. KEGG pathway analysis revealed changes in pathways associated with the metabolism of retinoids, steroid hormones and some long-chain polyunsaturated fatty acids (LCPUFA), such as linoleic and arachidonic acids. In addition, related KEGG terms suggested alterations of the immune system response. Beyond, with DEPs as seed nodes, a protein–protein interaction network was constructed using STRING software (https://string-db.org, Figure 2F). This analysis corroborated most of the terms offered by DAVID software (https://david.ncifcrf.gov, Figure 2G) but, in addition, revealed a tightly interconnected network around two central clusters. One was related to the fatty acid metabolism, but the other allowed us to highlight that disturbances in the metabolism of retinoids, steroids and LCPUFA kept a strong relationship. Disturbances in the cytochrome P450 were thereby identified as the underlying cause of these changes.

### 2.4. Genome-Wide Liver Methylation Changes for the F3 Generation Animals Ancestrally Exposed to Embryo Technologies

After quality assessment, three MOVET and four NC liver samples were kept in the final analysis. The mean number of raw reads was 92.8 ± 1.6 million, with a mean CG content (%) of 52.6 ± 0.13. The percentage of reads having a “Phred quality score” of over 30 (base call accuracy: 99.9%) was 95.5 ± 0.22%. Of the 7303 differentially methylated windows, 2570 (35.2%) were hypomethylated, and 4733 (64.8%) were hypermethylated. The global mean methylation counts per window was 15.5 ± 0.16 (61.1 ± 0.46%) and 12.3 ± 0.18 (38.9 ± 0.46%) for MOVET and NC and samples, respectively. PCA of the differentially methylated windows (Figure 3A) and HM clustering of differentially methylated genes (DMGs, Figure 3B) revealed a good level of clustering for MOVET and NC samples, grouping them according to their origin. The MA-plot, scatter plot in a logarithmic scale of fold changes (*y*-axis) versus the mean expression signal (*x*-axis), revealed high methylation differences between MOVET and NC samples (Figure 3C). The generally high methylation level in MOVET and NC samples suggests that the genome has experienced a substantial gain of methylation after embryo manipulation. However, the landscape of the methylome variation suggests differential patterns of change depending on the genome regions. Thereby, 6 and 10 chromosomes were hypomethylated and hypermethylated in MOVET animals compared to NC one (Figure 3D). Concordantly, the comparative epigenomic analysis revealed 121 DMGs, of which 43 (35.5%) were hypomethylated, and 78 (64.5%) were hypermethylated in MOVET animals compared to the NC ones (Figure 3E). From these DMGs, DAVID software recognizes 94 genes whose gene name, associated chromosome/scaffold and methylation difference (Δβ) were annotated in Appendix A. Functional analysis showed relevant associations between hypomethylated DMGs with dysregulations of cellular functions, the apoptotic process and the glycerolipid metabolism (Figure 3F). Meanwhile, hypermethylated genes were associated with cellular responses to DNA damage, cell ageing, apoptotic signaling, and transcription regulation (Figure 3F). DAVID GO term enrichment and KEGG pathway analysis of both hypomethylated and hypermethylated genes are shown in Appendix A. Furthermore, Panther pie charts constructed from DMGs (both hypo- and hypermethylated) enabled us to identify biological functions sensitive to the transgenerational epigenetic reprogramming process, such as metabolic and immune system processes (Figure 3G).

## 3. Discussion

We performed an analysis of DNA methylation profile in liver tissue in a cohort of F3 generation animals conceived by MOVET and compared findings with counterpart naturally conceived animals. Previously, our laboratory demonstrated that the effect of embryo technologies is associated with long-term and transgenerational phenotypic variation [19]. The current study extends these previous analyses, connecting the epigenome, proteome and metabolome to deeper understand the effects of embryonic manipulation.

Hence, untargeted metabolomics analyses revealed alterations in both semi-polar and non-polar metabolites in MOVET animals. In agreement with our findings, several studies described non-polar and semi-polar metabolism alteration after embryonic manipulation [10,20,21,22,23,24,25]. Globally, semi-polar metabolites were up-accumulated while non-polar metabolites were down-accumulated in MOVET animals compared to NC animals, suggesting a metabolic adaptation strategy [10,11,12,13,14,15,16,17,18,19,20,21,22,23,24,25]. Related to this, Feuer et al. observed that conception by IVF reduces growth, impairs glucose tolerance and impacts the serum and liver metabolome, affecting a wide range of compounds such as carbohydrates, amino acids, and, especially, lipids [10,23,24]. We have previously demonstrated that embryonic manipulation incurs both direct, intergenerational and transgenerational effects, diminishing the growth performance and impacting the hepatic metabolism [19,26,27]. The effects observed in F3 generation MOVET animals were light alterations in serum glucose levels, which also has been reported previously in the F1 generation [27]. Nevertheless, MOVET animals maintained glucose levels within a healthy range (75–150 mg/dL) [18]. This also has been described in humans in the F1 generation [22,28]. The targeted analysis particularly revealed higher relative affection of the non-polar metabolic pathways than the semi-polar ones. In agreement, increasing studies based on embryonic manipulation have described progenies with hepatic disorders in the lipid metabolism in fetal [29] and adult mice [10,21,24,30], as well as serum lipid profile deviations in mice and human [20,23,31,32] and alterations in the glycerolipid metabolism [18]. We recently found alterations in steroid profiles associated with embryo exposure to the transfer and cryopreservation-transfer procedures [24]. Here, we, therefore, focused on the unsaturated fatty acids biosynthetic pathway in depth. We found severe changes in liver metabolic profiles, with MOVET animals having metabolites that are highly downregulated compared to the NC animals, in agreement with a previous study in mice [30]. These compounds merit special attention, as it has been extensively reported that they are crucial for normal development and health in fetuses, newborns, and later stages [33,34,35,36]. Transgenerational alteration of these metabolites has been proposed as possible mechanisms to explain the phenotypic changes observed in animals derived from embryo technologies during three consecutive generations [19]. We found the downregulation of linoleic and arachidonic acids metabolites, which were also confirmed at the proteomic level, in agreement with Wang et al. [30]. In particular, we have shown that KEGG pathways of DEPs reported variations in the “linoleic acid metabolism” and “arachidonic acid metabolism” but also denoted changes in the “steroid hormone biosynthesis” and “retinol metabolism”, in agreement with a recent transcriptomic analysis [19]. Intriguingly, DEPs in these four terms were grouped in the protein–protein interaction network, revealing that they belong to cytochrome P450. The cytochrome P450 proteins catalyze many lipophilic compounds’ metabolism, including linoleic acid, arachidonic acid, retinol, and steroid hormones [37,38]. All these components have pivotal physiological functions per se and their derived metabolites, so perturbation of these compounds might have implications for optimal development [34,35,39,40]. Some compounds related to arachidonic acid metabolism (e.g., eicosanoids, phosphatidylcholines), which are crucial for normal cell function, growth and immunity [35,41], were down-accumulated in MOVET animals. Furthermore, it is worth noting that MOVET animals had slightly lower serum cholesterol levels which also has been reported in the F1 generation [27] and human studies [20,31,32]. However, cholesterol levels were within a healthy range (12–116 mg/dL) [18]. Steroid hormones are synthesized from cholesterol by members of the P450 and, after interacting with liver receptors, governs pathways related to lipid and glucose homeostasis, liver growth, body growth and immunity [38,40,42]. This evidence might contribute to explain, in part, the reduced growth performance and liver phenotypic changes previously observed [19,25]. The biosynthesis of active retinoid derivatives from retinol is crucial for many physiological processes, including embryonic development, postnatal growth, and immune responses [41,43]. Overall, the findings suggest that although results on the direct connection between the modulation of liver homeostasis by ART and epigenetic regulation are still minimal, the evidence that ART altered phenotypes suggests that ART is likely to be a player in epigenetic events regulating homeostasis in adulthood. In our experience, we have not found evidence that embryo manipulation has an overwhelmingly negative effect on health in the rabbit model [19,25,26,27,44,45,46,47,48,49,50], in line with the mouse model. However, the main limitation of our studies and others is that those who develop adverse health outcomes may not do so until later in life.

The potential for embryonic manipulation to affect hepatic methylation has also been assessed. We found evidence of a difference in ‘global’ DNA methylation between MOVET and NC animals. Of the biological process in which the 121 DMGs were involved, 82.3% coincided with those attributed to DEPs. Again, we identified that DMGs were related to lipid metabolism, regulation and signaling of the apoptotic process, cell ageing, DNA damage responses through p53 mediator and immune function. Cell cycle arrest at G1 in response to DNA damage can be induced by p53, promoting senescence or apoptosis, an essential mechanism for embryogenesis, organogenesis, differentiation and reprogramming [51]. In addition, our results show significant enrichment among the GO terms related to the G1 transition at the proteomic level. Further studies on this issue could pave the way to elucidating the underlying mechanisms of the physiological changes exhibited by MOVET animals. Moreover, differential methylation patterns were identified throughout 16 chromosomes, with a marked tendency to hypermethylation. Specifically, some chromosomes such as 1, 12 and 19 showed more significant methylation changes, which might indicate that some genes are altered preferentially than others [52]. A high correlation was observed with our previous transcriptomic approach [18], as differentially methylated chromosomes contained 70.8% of the differentially expressed transcripts, and 81.1% downregulated transcripts belonged to generally hypermethylated (silenced) chromosomes (Appendix A). The developmental time points at which embryo technologies are implemented are precisely when the large-scale reorganization of the epigenome takes place [14,52]. The dynamics of epigenome reprogramming have been widely explored [53,54,55]. However, further comprehensive epigenetic studies are required to compare embryos produced with different ARTs to gain insights into normal molecular regulation and correlate it with unperturbed embryonic and fetal development [8]. In this sense, the potential effect of embryonic manipulation on embryo epigenomes heads the list of mechanistic candidates that might explain the association between ART and its associated outcomes throughout life, including transgenerational ones [2,52,56,57,58].

Actually, blood hematological and biochemical parameters are critical for monitoring the disease diagnosis [59]. Hematological and serum biochemical responses of MOVET animals were within the normal physiological range for rabbit [18]; consequently, we can affirm that MOVET animals were seemingly healthy, with no evidence of adverse health outcomes with embryonic manipulation. Although molecular changes associated with the MOVET procedure were obvious, the potential health implications should not be over-interpreted given the lack of direct confirmation. To substantiate this finding, we previously published that MOVET animals have a similar fertilizing capacity to NC animals [19]. Hence, we suggest that the hepatic metabolic homeostasis of MOVET animals changed due to embryonic adaptive response [3,58]. Indeed, some of the metabolic changes described throughout this study are present in early embryos just after embryonic manipulation [11,15].

Some limitations are affecting this study: (i) the number of samples included are relatively small, which may restrict the interpretation of our results. (ii) We only tested one tissue. To know whether similar differences occur in other tissues, further studies are needed. (iii) This study was limited to males, which are less variable due to their constant hormone status. In females, the effects might likely be different [23], although this hypothesis requires further confirmation. To this end, future research works will be carried out to confirm and validate the biological relevance of these findings. (iv) Described approaches of molecular effects of ARTs have been performed on different omics landscapes, but integromics studies for possible overlaps between the results of different omics are required for predicting the risk for the use of ARTs or for predicting the course of the disease. Finally, we should keep in mind that ARTs are constantly evolving and rapidly applied without prior long-term safety evaluation to accommodate assisted reproduction needs [5]. Therefore, this study must encourage the development of new approaches to ensure the safest use of ART.

In conclusion, our studies have shown that embryo manipulation is associated with epigenetic variation evidenced by DNA methylation patterns in the F3 generation that was never exposed to manipulation. Furthermore, DNA methylation leads to hepatic metabolic homeostasis adjustments, which result in phenotypic plasticity without having an apparent negative impact on overall health. Thus, we provide evidence that epigenetic response to embryo manipulation plays a crucial role in the adaptive response. However, further research is needed to validate these findings, the biological significance and relevance, and fully understand embryo technologies’ mechanisms and marks. Besides, the relative extent to which epigenetic variation depends on each reproductive technique remains to be clarified.

## 4. Materials and Methods

### 4.1. Animals and Ethical Statements

Californian rabbits belonging to the Universitat Politècnica de València were used throughout the experiment [60]. The rabbit model provides some advantages over rodents in reproductive studies, as some reproductive biological processes exhibited by humans are more similar in rabbits than those in mice [61,62]. For example, humans and rabbits present a similar chronological embryonic genome activation, gastrulation and hemochorial placenta structure. In addition, using rabbits, it is possible to know the exact timing of fertilization and pregnancy stages due to their induced ovulation [62].

### 4.2. Experimental Design

The direct effect of the embryo manipulation was present in the embryos which formed the F1 generation and over the germline developing within these embryos, which ultimately created the F2 generation. Then, the F3 generation is the first not directly exposed to embryo manipulation effects [63]. With this aim, two experimental progenies were established as described before [19,27]. Briefly, one was generated from 158 vitrified embryos derived from 13 donors, which were transferred into 13 foster mothers (MOVET progeny). The other was constituted as a control progeny by naturally conceived animals (NC progeny), from 14 females inseminated contemporaneously with the previous donors. Embryo cryopreservation and transfer procedures have been described in detail previously [61]. At birth, 77 NC and 69 embryo manipulated animals were obtained. Both progenies were mated over two subsequent generations within each experimental group without any embryonic manipulation. To reduce the inbreeding, mating between animals with common grandparents was avoided. Hence, 61 NC and 56 MOVET animals were obtained for the F2 generation, and 61 NC and 64 MOVET animals constituted the F3 generation. Animals of both progenies in each generation were housed together in the same room throughout the experiment. After weaning in the 4th week, animals were caged collectively (8 rabbits per cage) until the 9th week. After that, animals were housed individually (flat deck indoor cages; 75 × 50 × 40 cm). In order to reduce confounding factors, the analysis was restricted to males, as they are thought to be less variable due to their constant hormone levels [64]. Once F3 progenies (MOVET and NC) reached the adult stage (56th week), hematological and biochemical analyses on peripheral blood were addressed to evaluate and compare their health status. After that, a comparative metabolomic, proteomic and epigenomic study was carried out. Figure 4 illustrates the experimental design.

### 4.3. Determination of Hematological and Biochemical Parameters on Peripheral Blood

Before euthanasia, 20 individual blood samples (10 MOVET and 10 NC) were taken from the central ear artery. Animals were selected randomly, keeping one animal of each litter (parity) within each experimental group. From each animal, two blood samples were taken. The first was dispensed into an EDTA-coated tube (Deltalab S.L., Barcelona, Spain) and the other into a serum-separator tube (Deltalab S.L., Barcelona, Spain). According to the manufacturer, blood count was performed from EDTA tubes at most 10 min after the collection using an automated veterinary hematology analyzer MS 4e automated cell counter (MeletSchloesing Laboratories, Osny, France) instructions. The blood parameters recorded were as follows: white blood cells, lymphocytes, monocytes, granulocytes, red blood cells, hemoglobin and hematocrit. From the second tube, biochemical analyses of the serum glucose, cholesterol, albumin, total bilirubin and bile acids were performed as hepatic metabolic indicators. Briefly, samples were immediately centrifuged at 3000× *g* for 10 min, and serum was stored at −20 °C until analysis. Then, glucose, cholesterol, albumin and total bilirubin levels were analyzed by enzymatic colorimetric methods, while bile acids were measured by photometry. All the methodologies were performed in an automatic chemistry analyzer model Spin 200E (Spinreact, Girona, Spain), following the manufacturer’s instructions. All samples were processed in duplicate. A general linear model (GLM) was fitted for the statistical analysis of the peripheral blood parameters, including as fixed effect the experimental group with two levels (MOVET and NC). Data were expressed as least-squares means ± standard error of means. Differences of *p*-value ≤ 0.05 were considered significant. Statistical analysis was performed with SPSS 21.0 software package (SPSS Inc., Chicago, IL, USA).

### 4.4. Sample Collection for Molecular Study

The uniformity of the liver tissue (four major cell types, of which hepatocytes constitute ≈70% of the total liver cell population) facilitates the sampling [65]. Then, individual liver samples were randomly taken from each animal (one rabbit, one sample). After that, samples were washed with phosphate-buffered saline solution, and each sample was divided into three parts. Two of them were directly flash-frozen in liquid nitrogen and stored at −80 °C for the metabolomic and proteomic study. The other was stored in RNA-later (Ambion Inc., Huntingdon, UK) at −20 °C for the epigenomic analysis. Finally, each sample for the metabolomic approach was generated by mixing tissues from 4 different animals (pools). Thus, 12 pooled samples (6 MOVET and 6 NC) were used for the metabolome study. Meanwhile, individualized samples were used for proteome and epigenome interrogation, which came from the same individuals used for our previous transcriptomic approach [19]. Then, 8 individual samples (4 MOVET and 4 NC) were used in the proteome and DNA methylation analysis.

### 4.5. Semi-Polar and Non-Polar Metabolome Analysis

Targeted and untargeted liquid chromatography–electrospray ionization–high resolution mass spectrometry (LC-ESI-HRMS) analysis of the hepatic semi-polar metabolome were performed as previously described [66,67,68], while targeted and untargeted liquid chromatography–atmospheric pressure chemical ionization–high resolution mass spectrometry (LC-APCI-HRMS) analyses of the non-polar metabolome were carried out as reported previously [69,70,71]. Untargeted metabolomics was performed using the SIEVE software (Thermofisher Scientific, San Francisco, CA, USA). Differentially accumulated metabolites (DAMs) were detected by a statistical analysis (one-way ANOVA plus Tukey’s pairwise comparison) using the SPSS software (SPSS Inc., Chicago, IL, USA), considering a *p*-value ≤ 0.05. Principal components analysis (PCA) and heat maps (HM) hierarchical clustering of untargeted metabolomes were performed using the ClustVis online software (https://biit.cs.ut.ee/clustvis/, accessed on 1 June 2021). Targeted metabolite identification was performed by comparing chromatographic and spectral properties with authentic standards (if available) and reference spectra, in house database, literature data, and based on the m/z accurate masses, as reported in the Pubchem database (http://pubchem.ncbi.nlm.nih.gov/, accessed on 1 June 2021) for monoisotopic mass identification, or on the Metabolomics Fiehn Lab Mass Spectrometry Adduct Calculator (http://fiehnlab.ucdavis.edu/staff/kind/Metabolomics/MS-Adduct-Calculator/, accessed on 1 June 2021) in the case of adduction detection. Finally, DAMs were detected as described above. Metabolites were quantified relatively by normalizing the internal standard (formononetin and DL-α-tocopherol acetate) amounts.

### 4.6. Comparative Proteomic Analysis

The proteome analyses were performed in the Proteomics Unit of the University of Valencia, Valencia, Spain (a member of the PRB2-ISCIII ProteoRed Proteomics Platform). The analysis of the hepatic proteome was performed as previously described [27]. First, we conducted a data-dependent acquisition (DDA) analysis to study the complete proteome by building up a spectral library using in-gel digestion and LC-MS/MS. Then, a sequential window acquisition of all theoretical fragment ion spectra mass spectrometry (LC-SWATH-MS) analysis was performed to determine quantitative differences in liver protein composition among the experimental rabbit progenies. For protein identification, validation and quantification, data were analyzed as previously described [27]. Briefly, library LC-MS/MS data were processed using the Paragon algorithm [72] of ProteinPilot search engine (AB SCIEX, Alcobendas, Madrid, Spain) and the Uniprot Mammalia protein sequence database (1,376,814 proteins searched). The Protein-Pilot Pro GroupTM Algorithm grouped identified proteins. The resulting Protein-Pilot group file was loaded into PeakView^®^ (AB SCIEX, Alcobendas, Madrid, Spain), and peaks from SWATH runs were extracted with a peptide confidence threshold of 95% confidence and an FDR < 0.01. After peptide detection, peptides were aligned among different samples using peptides detected at high confidence from the library. The extracted ion chromatograms were integrated, and the areas were used to calculate the total protein quantity. The mass spectrometry proteomics data were deposited with the ProteomeXchange Consortium via the PRIDE [73] partner repository with the dataset identifiers: PXD017972 (SWATH data) and PXD016874 (Spectral Library data).

The quantitative data obtained by PeakView^®^ were analyzed using MarkerView^®^ (v1.2, AB SCIEX, Alcobendas, Madrid, Spain). Normalization of the calculated areas was done by summing total areas. A t-test was used to identify the DEPs among the two experimental groups (VT and NC). Proteins were considered differentially expressed with an adjusted *p*-value ≤ 0.05. Rabbit (*O. cuniculus*) identifiers were obtained using the Blast tool from UniProt, keeping the output with the high-identity score. Principal components analysis (PCA) and heat map (HM) clustering were performed using ClustVis (https://biit.cs.ut.ee/clustvis/, accessed on 1 June 2021). Functional descriptive pie charts of DEPs were obtained from the Panther web tool (http://pantherdb.org/, accessed on 1 June 2021) using Homo sapiens as a reference and the human orthologous gene names (obtained from Biomart-Ensembl web tool: https://www.ensembl.org/info/data/biomart/index.html, accessed on 1 June 2021) as input data. Functional annotation of DEPs, enrichment analysis of their associated “Gene Ontology” (GO) terms and “Kyoto Encyclopedia of Genes and Genomes” (KEGG) pathways analysis were computed using the free available bioinformatics software DAVID Functional Annotation Tool (https://david.ncifcrf.gov/home.jsp, accessed on 1 June 2021; v6.8), considering a *p*-value (modified Fisher’s exact test, EASE score) of less than 0.05. In addition, DEPs were sent to the Search Tool for the Retrieval of Interacting Genes/Proteins (STRING; https://string-db.org/, accessed on 1 June 2021; v11.0) to build a functional protein association network.

### 4.7. Genome-Wide DNA Methylation Profiling by MBD-Seq

A double extraction of total DNA and RNA of the liver was extracted using Ambion (mirVana) and Qiagen (AllPrep) columns following the protocol of Peña-Llopis and Brugarolas [74]. RNA samples were used previously in our transcriptomic study [19]; meanwhile, DNA samples were used to perform the epigenetic approach. First, DNA integrity was checked using 1% agarose electrophoresis. Then, the MBD-seq service was provided by Macrogen Inc. (Seoul, Republic of Korea).

According to the manufacturer’s instructions, methylated DNA was obtained using the MethylMiner Methylated DNA Enrichment Kit (Invitrogen, Carlsbad, CA, USA). Briefly, 1 μg of genomic DNA fragmentation was performed using adaptive focused acoustic technology (AFA; Covaris) and captured by MBD proteins. The methylated DNA was eluted in the high-salt elution buffer. DNA in each eluted fraction was precipitated using glycogen, sodium acetate, and ethanol and resuspended in DNase-free water. The eluted DNA was used to generate libraries following the standard protocols of the TruSeq Nano DNA Library Prep kit (Illumina). The eluted DNA was repaired, an A was ligated to the 3′ end, and TruSeq adapters were ligated to the fragments. Once ligation was assessed, the adapter-ligated product was PCR amplified. The final purified product was quantified using qPCR according to the qPCR Quantification Protocol and qualified using Agilent Technologies 4200 TapeStation (Agilent technologies, Macrogen Inc, Seoul, Republic of South Korea). We sequenced using the HiSeqTM 2500 platform (Illumina, Macrogen Inc, Seoul, Republic of South Korea).

Reads were mapped using bwa-mem [75] against *O. cuniculus* genome assembly OryCun2.0.87 from ENSEMBL (https://www.ensembl.org/index.html, accessed on 1 June 2021). Only reads mapped against the genome with a mapq value of 57 or higher were kept using SAMtools [76]. For each sample, Reads Per Kilobase Million (RPKM) was calculated using MEDIPS [77]; these values were joined into a table and used for assessing how similar were the samples inside the experimental group by PCA visualization. The differentially methylated region (DMR) windows were calculated using Bioconductor package MEDIPS [77], using as arguments a window size of 250 bp, removing the excess of stacked reads in a given position with uniq = 10^−3^ and using as DMR detection method edgeR. DMR windows with an adjusted *p*-value < 0.005 were selected for further analyses. BED files were created from OryCun2.0.87 assembly’s annotation for 2 kbp adjacent gene regions, exon regions and intron regions with in-house scripts. Additionally, BED files for DMR results were created, merging adjacent windows. In order to check for DMR’s potential effects on gene expression, these BED files were compared to look for genes with DMR associated to them with BEDTools [78]. Once differentially methylated genes (DMGs) were identified, its functional annotation was performed using the DAVID Functional Annotation Tool as above.

## Figures and Tables

**Figure 1 ijms-22-09716-f001:**
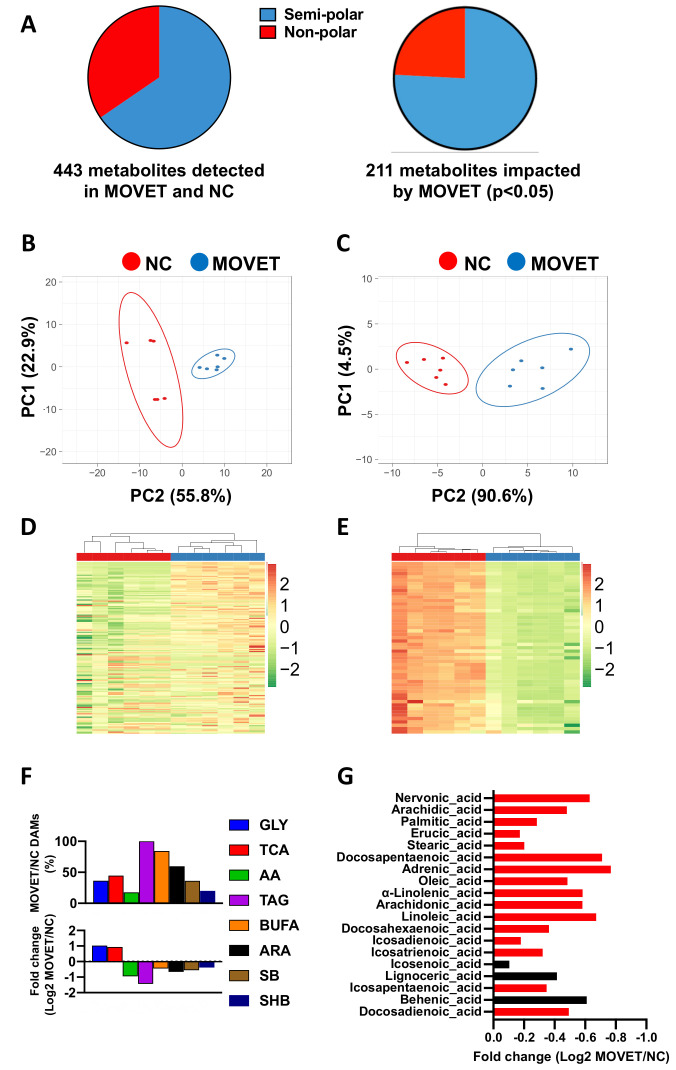
Metabolite profile changes in the liver of F3 generation animals derived from early embryos exposed to the combination of multiple ovulations, vitrification and embryo transfer (MOVET) and naturally conceived (NC) were compared. (**A**) Categorical distribution of the 443 metabolites detected by the untargeted analysis and the 211 differentially accumulated metabolites (DAMs) in VT rabbits. (**B**) Principal components analysis (PCA) of the untargeted semi-polar DAMs. (**C**) PCA of the untargeted non-polar DAMs. (**D**) Heat map (HM) clustering of the untargeted semi-polar DAMs. (**E**) HM clustering of the untargeted non-polar DAMs. (**F**) Bar graph showed the percentage of DAMs of MOVET versus NC comparison in several metabolic pathways (up) and the average fold change between the MOVET and NC group (down). GLY: glycolysis–gluconeogenesis; TCA: citric acid cycle; AA: biosynthesis of amino acids; TAG: triglycerides; BUFA: biosynthesis of unsaturated fatty acids; ARA: arachidonic acid metabolism; SB: steroid biosynthesis; SHB: steroid hormone biosynthesis. (**G**) Bar graph showed the average fold change of DAMs between MOVET and NC group in the biosynthesis of unsaturated fatty acids pathway. Red bars denote statistical differences. Black bars denote no statistical differences.

**Figure 2 ijms-22-09716-f002:**
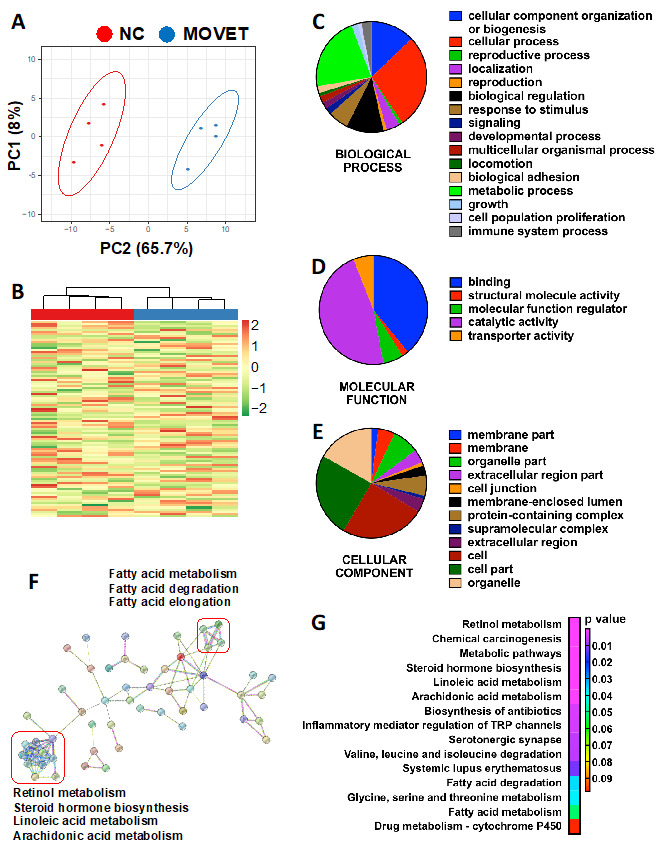
Protein profile changes in the liver of F3 generation animals derived from early embryos exposed to the combination of multiple ovulations, vitrification and embryo transfer (MOVET) and naturally conceived (NC) were compared. (**A**) Principal components analysis of the differentially expressed proteins (DEPs). (**B**) Heat map clustering of the DEPs. (**C**–**E**) Pie charts representing the DEPs distribution according to their biological process, molecular function, and cellular component. (**F**) Protein–protein interaction network of DEPs. Not interconnected DEPs were excluded. (**G**) KEGG analysis for the DEPs.

**Figure 3 ijms-22-09716-f003:**
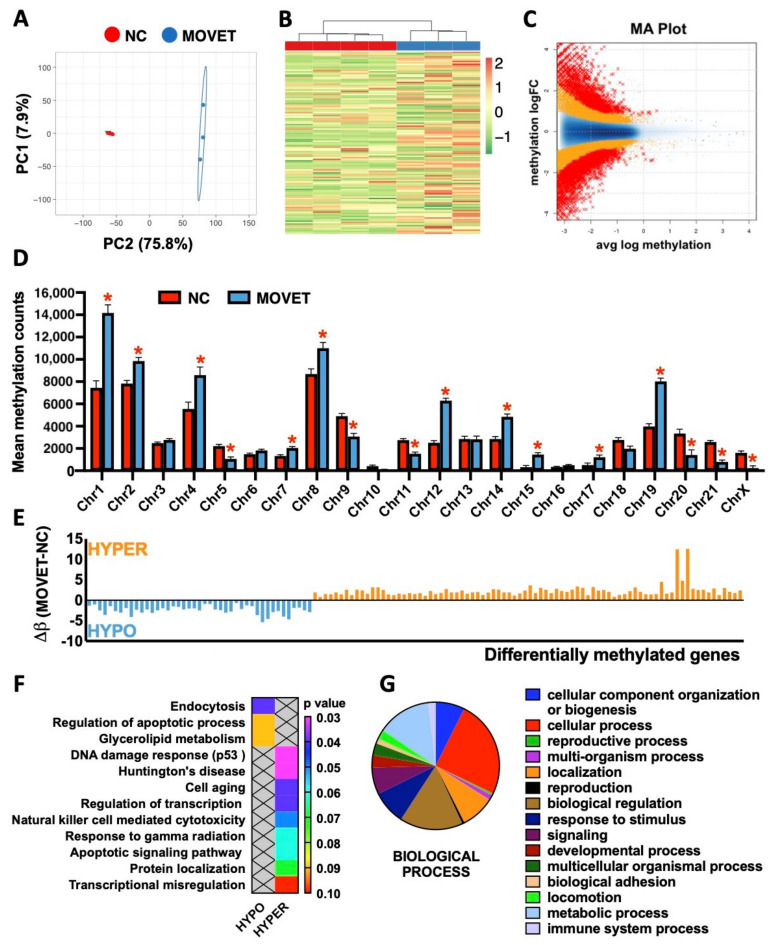
Genome-wide methylation changes in the liver of F3 generation animals derived from early embryos exposed to the combination of multiple ovulations, vitrification and embryo transfer (MOVET) and naturally conceived (NC) were compared. (**A**) Principal components analysis of the differentially methylated windows. (**B**) Heat map clustering of the differentially methylated genes (DMGs). (**C**) MA plot representing data comparison between MOVET and NC samples. (**D**) Total number of methylation counts per chromosome. Asterisks denote statistical differences. (**E**) Methylation difference (Δβ) of the 121 DMGs calculated as mean MOVET DNA methylation minus mean NC DNA methylation. (**F**) Gene Ontology (biological process) and KEGG analysis for the hyper- and hypo-DMGs of MOVET versus NC. (**G**) Pie chart representing the total DMGs distribution according to their biological process.

**Figure 4 ijms-22-09716-f004:**
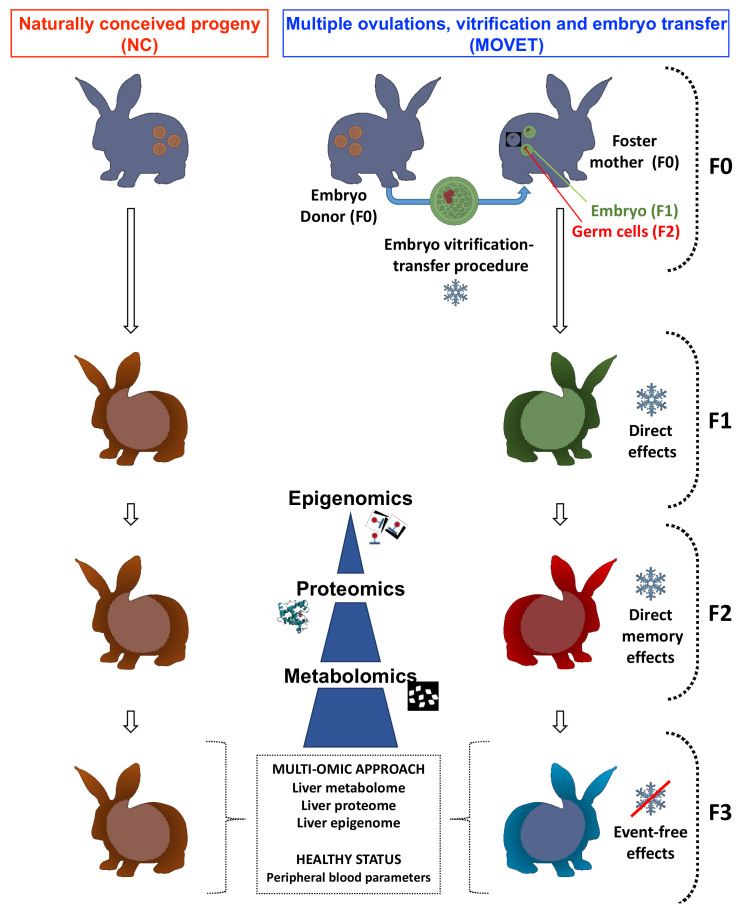
The experimental design used to examine DNA methylation meets proteomics and metabolomics in the liver tissue after embryo manipulation. The experimental progenies were compared in the F3 generation animals to exclude a direct effect of the stimulus on the embryos’ somatic cells and germ cells.

**Table 1 ijms-22-09716-t001:** Comparison of hematological and biochemical profile changes of F3 generation animals derived from early embryos exposed to the combination of multiple ovulations, vitrification and embryo transfer (MOVET) and naturally conceived (NC).

Blood Parameters	NC	MOVET
**Hematology**		
**White blood cells** (10^3^/mm^3^)	9.1 ± 1.71	7.4 ± 1.71
**Lymphocytes** (%)	40.1 ± 2.75	36.0 ± 2.75
**Monocytes** (%)	7.3 ± 0.72	6.4 ± 0.72
**Granulocytes** (%)	46.7 ± 2.83	49.3 ± 2.83
**Red blood cells** (10^6^/mm^3^)	6.1 ± 0.27	6.3 ± 0.27
**Hemoglobin** (g/dL)	12.6 ± 0.63	13.1 ± 0.63
**Hematocrit** (%)	43.8 ± 2.26	44.0 ± 2.26
**Serum metabolites** ^+^		
**Albumin** (g/dL)	4.4 ± 0.91	4.5 ± 0.91
**Bile acids** (µmol/L)	6.9 ± 1.08	7.1 ± 1.08
**Cholesterol** (mg/dL)	32.2 ± 1.94 ^a^	25.6 ± 1.94 ^b^
**Glucose** (mg/dL)	103.3 ± 10.39 ^b^	138.7 ± 10.39 ^a^
**Bilirubin total** (mg/dL)	0.1 ± 0.02	0.1 ± 0.02
**Number of animals**	10	10

Data are expressed as least-square means ± standard error of means. ^a,b^ Values in the same row with different superscript are significantly different (*p* < 0.05). ^+^ Serum indicators of hepatic function.

## Data Availability

The datasets used and/or analyzed during the current study are available from the corresponding author on reasonable request.

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
