# Peer review of "Early Embryo Exposure to Assisted Reproductive Manipulation Induced Subtle Changes in Liver Epigenetics with No Apparent Negative Health Consequences in Rabbit"

_ijms, 2021, doi:10.3390/ijms22189716_

Round 1
Reviewer 1 Report
In the present study, the authors analyzed the long-term effect of ART (multiple ovulation, vitrification and embryo transfer - MOVET) on hepatic metabolism and DNA methylation in the F3 generation. The authors found that MOVET animals presented altered metabolic, proteomic and epigenomic differences, especially regarding lipid metabolism. The author concluded that ART induced long-term effects on hepatic tissue metabolic function, associated with DNA methylation variations, although no apparent negative impact was found on animal health. In my opinion, the article is interesting, timely and well written. The objectives of the study are clearly stated, and the experimental design and results support the conclusions. The present study may be the basis for further research.
Specific comments:
- I would change the title since, based on the results, it is not clear to me whether this is a tissue adaptation for homeostasis or if the differences found are consequences of failures in the epigenetic reprogramming of the ART-derived embryos.
- I suggest the abstract to be rewritten. Note that this section is sometimes difficult to understand. Some specific points: at the first sentence, concepts as “developmental programming”, embryonic plasticity”, phenotypic variations” are used but the relation among them is confused.
Line 14 – which environmental conditions?
Line 15 – I suggest “produced”, not “exposed”
Line 14-15 – Rewrite the sentence
Line 20 – Again, it is not clear to me that this is an adaptive variation, or simply the consequences of failures in epigenetic reprogramming that may have occurred previously in MOVET embryos.
Introduction
This section is well written nevertheless I suggest the authors to include more information regarding the epigenetic reprogramming of early embryos and how ART may affect that.
Results
Legends, in general, may be improved. For instance, it is not clear in Figure 1F and 1G in which group these pathways/metabolites are up or down represented.
Section 2.3 and supplementary material 2 and 3 – I suggest the authors to present these data describing DEPs and enriched pathways up and down regulated in each group. This information is not clear.
Section 2.4 – line 164-165 – in this section it is also important to better describe hypo and hypermethylated regions or DMGs for each group.
Line 178 – The author found 121 DMGs but do not give any information about other genomic regions differentially methylated. The authors also reference a recent publication in which they described the transcriptional pattern of the same tissue. I suggest the authors to check in these data if these DMGs are also DEGs and explore this integrative analysis.
Discussion
Line 212-213 – Could you explain in more detail this “compensatory mechanism”?
Line 262-263 – Please, explain better this statement.
Line 285-286 – In my opinion this information is the key to the work and should be better explored.
Material and Methods
Why did the authors analyze proteomic and DNA methylation in individual samples and pooled tissues for metabolomic analysis?
Author Response
Specific comments:
- I would change the title since, based on the results, it is not clear to me whether this is a tissue adaptation for homeostasis or if the differences found are consequences of failures in the epigenetic reprogramming of the ART-derived embryos.
According to the suggestion, the title has been changed to “Early embryo exposure to assisted reproductive manipulation induced subtle changes in liver epigenetics with no apparent negative health consequences in rabbit”
- I suggest the abstract to be rewritten. Note that this section is sometimes difficult to understand. Some specific points: at the first sentence, concepts as “developmental programming”, embryonic plasticity”, phenotypic variations” are used but the relation among them is confused.
Line 14 – which environmental conditions?
Line 15 – I suggest “produced”, not “exposed”
Line 14-15 – Rewrite the sentence
The abstract has been rewritten to clarify the concepts according to the reviewer indications.
Line 20 – Again, it is not clear to me that this is an adaptive variation, or simply the consequences of failures in epigenetic reprogramming that may have occurred previously in MOVET embryos.
As we indicated previously, the abstract had been rewritten to clarify the concepts according to the reviewer's indications. In this sense, the use of the concept of adaptative variation has been minimised based on the reviewer's indication that failures in epigenetic reprogramming also could be the result of embryo manipulation.
Introduction
This section is well written nevertheless I suggest the authors to include more information regarding the epigenetic reprogramming of early embryos and how ART may affect that.
The second paragraph of the introduction has been rewritten following the guidance of the reviewer comment.
Results
Legends, in general, may be improved. For instance, it is not clear in Figure 1F and 1G in which group these pathways/metabolites are up or down represented.
The figure legends have been improved to facilitate the understanding of the figures. In this study, a direct comparison was made between the two experimental; MOVET versus NC. The average fold change between the MOVET and NC group (Log2 MOVET/NC) indicates that the positive sign denotes up-accumulated metabolites in the MOVET group while the negative denotes down-accumulated in the MOVET group. Figure 1 F and G have been modified to designate that the Fold change average was between MOVET/NC. That, in Figure, 1F the GLY: Glycolysis- gluconeogenesis and TCA: Citric acid cycle pathways are up-accumulated in the MOVET group, while the others pathways down-accumulated. In Figure 1G, all the metabolites in the biosynthesis of unsaturated fatty acids are down-accumulated in the MOVET group.
Section 2.3 and supplementary material 2 and 3 – I suggest the authors to present these data describing DEPs and enriched pathways up and down regulated in each group. This information is not clear.
As previously we have indicated that comparisons were done between the MOVET versus NC group. So, the negative values are down (accumulated, expressed, or hypomethylated) in the MOVET group, while the positive are up. The comparison direction has been included in all the titles of the supplementary tables and the figure legends.
Section 2.4 – line 164-165 – in this section it is also important to better describe hypo and hypermethylated regions or DMGs for each group.
This information is included in the line 181-183.
Line 178 – The author found 121 DMGs but do not give any information about other genomic regions differentially methylated. The authors also reference a recent publication in which they described the transcriptional pattern of the same tissue. I suggest the authors to check in these data if these DMGs are also DEGs and explore this integrative analysis.
The epigenetic computer analysis was commissioned 2 years ago, and at that time, the study was not contracted to find out the genomic regions differentially methylated. This information will be incorporated into an analytic study to identify biomarkers when we can address an integrated analysis. Nevertheless, a high correlation was observed with our previous transcriptomic approach (Garcia-Dominguez, X.; Marco-Jiménez, F.; Peñaranda, D.S.; Diretto, G.; García-Carpintero, V.; Cañizares, J.; Vicente, J.S. Long-term and transgenerational phenotypic, transcriptional and metabolic effects in rabbit males born following vitrified embryo transfer. Sci. Rep. 2020, 10, doi:10.1038/s41598-020-68195-9), as differentially methylated chromosomes contained 70.8% of the differentially expressed transcripts, and 81.1% downregulated transcripts belonged to generally hypermethylated (silenced) chromosomes (Supplementary Figure 1). This information and the corresponding figure have been added to the discussion section.
Discussion
Line 212-213 – Could you explain in more detail this “compensatory mechanism”?
Maybe the term "compensatory mechanism" was not precise. Instead, we hypothesise that there is a metabolic adaptation strategy. We recognise the concept of adaptation strategy as a mechanism to modify the metabolites values without these incurring health problems, as observed in the animals through the generations of our rabbit model. In line with our results, Feuer et al. observed that conception by IVF reduces growth, impairs glucose tolerance, impacts the serum and liver metabolome, producing a readjustment in a wide range of components. This sentence has been rewritten following the reviewer's comment.
Line 262-263 – Please, explain better this statement.
The phrase has been rewritten.
Line 285-286 – In my opinion this information is the key to the work and should be better explored.
The phrase has been rewritten to show that it is a reprogramming period of the epigenome, well known in mice and humans, but that little has been related to ART. We try to avoid being extremely speculative based on the current literature.
Material and Methods
Why did the authors analyze proteomic and DNA methylation in individual samples and pooled tissues for metabolomic analysis?
Based on our laboratory logistics, the metabolome analysis was the last one performed, and due to that metabolomics is a powerful approach because metabolites and their concentrations, unlike other “omics” measures, directly reflect the underlying biochemical activity and state of the tissue; we decided to use a pool of animals to have greater power in our results. To this end, we used 48 animals (24 animals per experimental group) in 12 pools with 4 animals per pool. The soundness of this descriptive study is that there are indicators that suggest that ART is likely to modify lipid metabolism in adulthood at all levels (connecting the epigenome, transcriptome, proteome and metabolome).
Reviewer 2 Report
In this paper, Garcia-Dominguez, et al. report that embryo manipulation induces adaptive changes in the hepatic tissue linked to variation in metabolites, and these are possibly derived from alteration in DNA methylation status. The reviewer acknowledges that the questions that the authors are tackling are interesting and of great importance. However, the paper lacks substantial amount of evidence to support the authors’ claims.
Major comments
- What is the main reason(s) for using rabbit for this study? Please describe briefly in the main text, the reason(s), and the advantage(s) of using rabbit as a model animal for conducting their study. Can the authors state that rabbit is an optimal model animal to study this kind of question?
- As authors admit in line 302, the number of samples analyzed are small, and the reviewer finds this critical drawback to support their claims.
- How can the authors distinguish between genetical background differences with MOVET? How can the authors state that the differences observed in Figure 1 in MOVET are indeed occurring from embryo manipulation?
- It is not clear, whether alterations in DNA methylation status are really affecting expression profiles of metabolites in Figure 1. Do these DNA methylation changes (hyper- and hypo-methylation) affect transcriptional levels of genes related to metabolisms? Moreover, the authors need to define where in the genome (promoter, enhancer, gene body, etc.), these changes are occurring.
- As these changes in metabolites did not affect overall health condition of MOVET, how can the authors state that these changes are important? The reviewer finds the title to be overstated.
- Why are the number of “standard error of means” in Table 1, exactly the same between NC and MOVET?
Author Response
Major comments
- What is the main reason(s) for using rabbit for this study? Please describe briefly in the main text, the reason(s), and the advantage(s) of using rabbit as a model animal for conducting their study. Can the authors state that rabbit is an optimal model animal to study this kind of question?
The rabbit model provides some advantages over rodents in reproductive studies, as some reproductive biological processes exhibited by humans are more similar in rabbits than those in mice (Fischer et al., 2012;Garcia-Dominguez et al., 2019). For example, humans and rabbits present a similar chronological embryonic genome activation, gastrulation and hemochorial placenta structure. In addition, using rabbits, it is possible to know the exact timing of fertilisation and pregnancy stages due to their induced ovulation (Fischer et al., 2012). This paragraph has been included in the main text (Line 333-338).
Garcia-Dominguez X, Marco-Jimenez F, Viudes-de-Castro MP, Vicente JS. Minimally Invasive Embryo Transfer and Embryo Vitrification at the Optimal Embryo Stage in Rabbit Model. J Vis Exp. 2019 May 16;(147). doi: 10.3791/58055.
Fischer B,Chavatte-Palmer P,Viebahn C,NavarreteSantos A,Duranthon V. Rabbit as a reproductive model for human health. Reproduction. 2012;144:1–10.
- As authors admit in line 302, the number of samples analyzed are small, and the reviewer finds this critical drawback to support their claims.
We know that the number of samples analyzed for epigenome and proteome are relatively low, so we indicated it in the main text. However, based on our laboratory logistics, the metabolome analysis was the last one performed, and due to that metabolomics is a powerful approach because metabolites and their concentrations, unlike other “omics” measures, directly reflect the underlying biochemical activity and state of the tissue; we decided to use a pool of animals to have greater power in our results. To this end, we used 48 animals (24 animals per experimental group) in 12 pools with 4 animals per pool (this is indicated in the main text Line X to X). Accordingly, we understand that the metabolome results are robust, and finding connections with the proteome and the epigenome is not by chance. Moreover, similar results have been recently described in the mouse model (Narapareddy L, Rhon-Calderon EA, Vrooman LA, Baeza J, Nguyen DK, Mesaros C, Lan Y, Garcia BA, Schultz RM, Bartolomei MS. Sex-specific effects of in vitro fertilization on adult metabolic outcomes and hepatic transcriptome and proteome in mouse. FASEB J. 2021 Apr;35(4):e21523. doi: 10.1096/fj.202002744R). The soundness of this descriptive study is that there are indicators that suggest that ART is likely to modify lipid metabolism in adulthood at all levels (connecting the epigenome, transcriptome, proteome and metabolome).
- How can the authors distinguish between genetical background differences with MOVET? How can the authors state that the differences observed in Figure 1 in MOVET are indeed occurring from embryo manipulation?
Our experimental design allows the genetic effect to be minimized. Our animals belonging to the Universitat Politècnica de València are a closed population. This animal comes from the fusion of two paternal lines, one founded in 1976 with Californian rabbits reared by Valencian farmers and another founded in 1981 with rabbits belonging to specialized paternal lines (Estany J, Camacho J, Baselga M, Blasco A. Selection response of growth rate in rabbits for meat production. Genet Sel Evol. 1992;24: 527–537.) The current generation is the 39th. The size of this line is around 120 does and 25 males. This number of males is enough for breeding with keeping the inbreeding coefficient at a low level. Matings between relatives sharing a grandparent are avoided, and each male contributed at least one offspring to the next generation. Selection is conducted in non-overlapping generations, and the generation interval is around 10 months. Based on this population, we generated 2 populations to develop this study (MOVET and NC). Both populations had the same sire families (n=10). Moreover, samples were obtained from the animals with the same sire families to avoid biases as much as possible. Given that both populations were created simultaneously and maintained throughout the entire experiment in the same room of the animal facility, we find that the only difference between both populations is embryo manipulation. Furthermore, all the environmental effect has been the same (temperature, light exposure, feed, animal handling, etc.)
- It is not clear, whether alterations in DNA methylation status are really affecting expression profiles of metabolites in Figure 1. Do these DNA methylation changes (hyper- and hypo-methylation) affect transcriptional levels of genes related to metabolisms? Moreover, the authors need to define where in the genome (promoter, enhancer, gene body, etc.), these changes are occurring.
A high correlation was observed with our previous transcriptomic approach (Garcia-Dominguez, X.; Marco-Jiménez, F.; Peñaranda, D.S.; Diretto, G.; García-Carpintero, V.; Cañizares, J.; Vicente, J.S. Long-term and transgenerational phenotypic, transcriptional and metabolic effects in rabbit males born following vitrified embryo transfer. Sci. Rep. 2020, 10, doi:10.1038/s41598-020-68195-9), as differentially methylated chromosomes contained 70.8% of the differentially expressed transcripts, and 81.1% downregulated transcripts belonged to generally hypermethylated (silenced) chromosomes (Supplementary Figure 1). This information and the corresponding figure have been added to the discussion section.
The epigenetic computer analysis was commissioned 2 years ago, and at that time, the study was not contracted to find out the genomic regions differentially methylated. This information will be incorporated into an analytic study to identify biomarkers when we can address an integrated analysis.
- As these changes in metabolites did not affect overall health condition of MOVET, how can the authors state that these changes are important? The reviewer finds the title to be overstated.
We agree with this comment that moreover is in line with reviewer 1. Based on this, we propose the title “Early embryo exposure to assisted reproductive manipulation induced subtle changes in liver epigenetics with no apparent negative health consequences in rabbit”
- Why are the number of “standard error of means” in Table 1, exactly the same between NC and MOVET?
The means have the same standard error because both groups have an equal number of cases (n=10).
Reviewer 3 Report
The article entitled “Epigenetic variation to adapt hepatic lipid homeostasis in response to embryo manipulation in rabbit” is well written and executed. However there are some concerns regarding the study.
- This study mainly focused on the effect of epigenetic variations on the hepatic function. But basic liver function tests are not done such as Alanine transaminase (ALT), Aspartate transaminase (AST), Alkaline phosphatase (ALP), Albumin and total protein, Bilirubin, Gamma-glutamyltransferase (GGT), L-lactate dehydrogenase (LD), Prothrombin time (PT).
- The effect of multiple ovulation, vitrification and embryo transfer must be studied individually, so as to improve ART (Assisted Reproductive techniques).
- This study is more descriptive, still some functional correlation is needed
Author Response
The article entitled “Epigenetic variation to adapt hepatic lipid homeostasis in response to embryo manipulation in rabbit” is well written and executed. However there are some concerns regarding the study.
- This study mainly focused on the effect of epigenetic variations on the hepatic function. But basic liver function tests are not done such as Alanine transaminase (ALT), Aspartate transaminase (AST), Alkaline phosphatase (ALP), Albumin and total protein, Bilirubin, Gamma-glutamyltransferase (GGT), L-lactate dehydrogenase (LD), Prothrombin time (PT).
In this study, firstly, we aimed to carry out a basic test of health and liver functionality. Since none of the values ​​indicating health alterations (all parameters ranged between normal values), no additional serological tests were planned. Furthermore, we understood that our liver metabolome study based on 153 metabolites would be more consistent than extending other serological tests. Admittedly, it is fitting that more tests could have been done, providing more robust information. If the reviewer considers it imperative, these tests could be performed.
- The effect of multiple ovulation, vitrification and embryo transfer must be studied individually, so as to improve ART (Assisted Reproductive techniques).
We fully understand the reviewer's approach, as this was one of the group's discussions before starting our research line in 2015. Currently, our approach is not to know the contribution of each of these effects but to consider the collective procedure needed to generate offspring. Nevertheless, in some of our previous studies, we evaluated these effects separately since we are also interested in knowing how each of the stages of the process intervenes (Garcia-Dominguez X, Diretto G, Frusciante S, Vicente JS, Marco-Jiménez F. Metabolomics Analysis Reveals Changes in Preimplantation Embryos Following Fresh or Vitrified Transfer. Int J Mol Sci. 2020 Sep 26;21(19):7116). We understand that the most appropriate experimental design depends on the objective of the study is. The approach is more complex due that interactions between reproductive techniques appear. For example, how to evaluate the effect of superovulation? When applying a superovulation treatment, the embryos must be manipulated to recover and transfer to a foster mother. Therefore, which is the control group? It would be necessary to generate two control groups, non-superovulated animals and non-superovulated and transferred animals. If we now add extra ART such as vitrification to the model, the number of experimental groups would grow notably to know the individual effects. If now we contemplate the minimum number of animals to avoid inbreeding throughout the generations in rabbits, the reviewer could understand that it is impossible to carry out this type of study in a conventional animal facility. Briefly, in rabbits, a minimum of 10 sire families are required in which crosses are done avoided common grandparents to guaranteeing that the inbreeding is less than 1% in each generation. It is equivalent to a minimum population size of around 60 animals. Through this brief history, we decided to carry out our rabbit model based on only 2 populations: a population that involves the vitrification technique (from multiple ovulation, vitrification and embryo transfer MOVET) to generate offspring compared to naturally-conceived (NC).
- This study is more descriptive, still some functional correlation is needed
We are very aware of the study's limitations and that it is a descriptive study since not all the omic studies have been integrated. Based on ARTs, no studies have been reported integrating the results of different omics landscapes. The value of our study related to published to date is that it shows that three generations after embryo manipulation, there are changes in liver functionality, although these are within the homeostatic balance. In addition, a new limitation commentary has been incorporated into the text, suggesting the necessity of the development of integromics studies. The new paragraph included the following phrase: (iv) Described approaches of molecular effects of ARTs had been performed on different omics landscapes, but integromics studies for possible overlaps between the results of different omics are required for predicting the risk for the use of ARTs or for predicting the course of the disease was identified. Moreover, a high correlation was observed with our previous transcriptomic approach (Garcia-Dominguez, X.; Marco-Jiménez, F.; Peñaranda, D.S.; Diretto, G.; García-Carpintero, V.; Cañizares, J.; Vicente, J.S. Long-term and transgenerational phenotypic, transcriptional and metabolic effects in rabbit males born following vitrified embryo transfer. Sci. Rep. 2020, 10, doi:10.1038/s41598-020-68195-9), as differentially methylated chromosomes contained 70.8% of the differentially expressed transcripts, and 81.1% downregulated transcripts belonged to generally hypermethylated (silenced) chromosomes (Supplementary Figure 1). This phrase and the corresponding figure have been added.
Round 2
Reviewer 2 Report
I have no further comment.
Reviewer 3 Report
Authors reviewed the paper thoroughly, answered all the queries